# EXercise to Prevent frailty and Loss Of independence in insulin treated older people with DiabetEs (EXPLODE): protocol for a feasibility randomised controlled trial (RCT)

Rachel Stocker ,[1] James Shaw,[1] Guy S Taylor,[1] Miles D Witham ,[2] Daniel J West[1]

[1]Population Health Sciences Institute, Newcastle University, Newcastle upon Tyne, UK
[2]NIHR Newcastle Biomedical Research Centre, Newcastle University, Newcastle upon Tyne, UK

**Correspondence to**
Dr Rachel Stocker;
rachel.stocker@newcastle.ac.
uk and
Dr Daniel J West;
daniel.west@newcastle.ac.uk

## ABSTRACT

**Introduction** There are 3.9 million people in the UK with diabetes. Sarcopenia, increased frailty and loss of independence are often unappreciated complications of diabetes. Resistance exercise shows promise in reducing these complications in older adult diabetes patients. The aim of this feasibility randomised controlled trial is to (1) characterise the physical function, cardiovascular health and the health and well-being of older adults with mild frailty with/without diabetes treated with insulin, (2) to understand the feasibility and acceptability of a 4-week resistance exercise training programme in improving these parameters for those with diabetes and (3) to test the feasibility of recruiting and randomising the diabetic participant group to a trial of resistance training.

**Methods and analysis** Thirty adults aged ≥60 years with insulin-treated diabetes mellitus (type 1 or 2), and 30 without, all with mild frailty (3–4 on the Rockwood Frailty Scale) will be recruited. All will complete blood, cardiovascular and physical function testing. Only the diabetic group will then proceed into the trial itself. They will be randomised 1:1 to a 4-week semisupervised resistance training programme, designed to increase muscle mass and strength, or to usual care, defined as their regular physical activity, for 4 weeks. This group will then repeat testing. Primary outcomes include recruitment rate, attrition rate, intervention fidelity and acceptability, and adherence to the training programme. A subset of participants will be interviewed before and after the training programme to understand experiences of resistance training, impact on health and living with diabetes (where relevant) as they have aged. Analyses will include descriptive statistics and qualitative thematic analysis.

**Ethics and dissemination** The North East-Newcastle and North Tyneside 2 Research Ethics Committee (20/ NE/0178) approved the study. Outputs will include feasibility data to support funding applications for a future definitive trial, conference and patient and public involvement presentations, and peer-reviewed publications.

**Trial registration number** ISRCTN13193281.

## Strengths and limitations of this study

► This is a novel study using mixed methods to examine the feasibility of carrying out a larger trial of a gym-based resistance exercise training programme with older, mildly frail adults with insulin-treated diabetes.

► A series of in-depth qualitative interviews will generate a better understanding of the barriers and facilitators to resistance-based exercise for this important patient group, where early intervention is key.

► This study is not intended to be a definitive trial, however, a mixed-methods approach will explore the impact of the training programme on various clinical outcomes and exercise experiences.

► As the study is limited to the North East of England, the sociodemographic characteristics of participants may differ to the population of the wider UK and other Western countries.

## INTRODUCTION
### Background

There are around 425 million people with diabetes worldwide, and by 2040 it is predicted that 1 in 10 people globally with have diabetes.[1] In the UK, there are around 4 million people living with diabetes[2] (90% type 2, 8% type 1, 2% other) with a further 1 million with undiagnosed diabetes. The prevalence of diabetes increases sharply with age, with 17.4% of those aged over 65 having diabetes, compared with 2% of those aged 16 to 44 in England.[3] All of those with type 1 diabetes require insulin treatment, and most of those with type 2 diabetes will eventually also require insulin treatment.[4 5] Diabetes represents >10% of the National Health Service (NHS) budget in direct treatment costs.[6] Modern advances in diabetes treatment

mean that people with diabetes are living longer,[7] even with the presence of the microvascular and macrovascular disease associated with long-term diabetes.

With an ageing population of people living with diabetes, it is important that strategies for improving both health, quality of life, and reducing treatment burden are identified. Long-term insulin treatment for diabetes is, however, associated with detrimental effects to health including hypoglycaemia and weight gain,[8] and may adversely affect muscle health.[9] Sarcopenia in particular leads to an increased risk of frailty, falls, physical disability, chronic metabolic disease and mortality.[10 11] The prevalence of frailty in older people with diabetes has been reported as ~32%–48%, which is significantly higher than that of 5%–10% in older persons without diabetes.[12 13] Recent studies have also demonstrated an increased risk of osteoporosis and fracture in older people with diabetes, compared with age matched non-diabetes controls.[14 15] Thus long-term diabetes and long-term insulin treatment, hypoglycaemia and age related physical decline, may carry an additional burden for those living with insulin treated diabetes in later life.

Physical activity and exercise interventions have been shown to improve outcomes associated with frailty and sarcopenia (such as muscle mass, muscle force production, cardiorespiratory fitness).[16] However, these interventions are not straightforward in people with insulin treated diabetes due to the risk of hypoglycaemia.[17] Aerobic exercise increases insulin sensitivity, changes the absorption and action of the injected insulin and increases metabolic rate dramatically.[18 19] Under these conditions, in a person without diabetes, the insulin concentrations would drop dramatically during and after exercise, however, in those with diabetes this is not possible (as the insulin is injected), and thus increases the risk of dangerously low blood glucose occurring. Nearly two-thirds of those with type 1 diabetes do not engage in any physical activity,[20] commonly due to fear of exercise-induced hypoglycaemia.[21]

Established insulin-treated non-obese type 2 diabetes shares many characteristics with type 1 diabetes, due to relatively greater insulin deficiency and lower insulin resistance than in type 2 diabetes associated with obesity. This includes intrinsic glucose variability with higher risk of impaired awareness of hypoglycaemia,[22 23] including severe events requiring assistance from others in treatment.[24] We hypothesise that mild frailty may have a comparable impact in type 1 diabetes and insulin-treated type 2 diabetes where body mass index (BMI) is <30 kg/m$^2$, with potentially comparable impacts of resistance exercise training.[25]

Resistance-based exercise, that is, repeated intense muscle contractions of isolated parts of the body against a fixed load, is associated with less fluctuation in blood glucose than aerobic exercise.[26–28] Resistance exercise is a potent stimulus for improving (1) muscle mass, (2) muscle strength and power, (3) bone health and (4) physical function, such as stair climbing.[29 30] Existing studies have shown resistance training to be a useful exercise modality in older, non-frail type 1 and type 2 diabetes patients.[31–33] It is also the modality of exercise with the most evidence for improving outcomes in older people with sarcopenia or frailty,[34 35] and generally well tolerated by this group.[36] However, data for its use in older people with diabetes are scant. As resistance training potentially carries less risk of blood glucose fluctuation to those with diabetes, due to differing hormonal responses to aerobic exercise, it has potential to be a preferred modality of exercise for this group, and may help sustain long-term engagement.[26–28]

Given the increasing ageing diabetes population, and the increased risk of sarcopenia and frailty in this group, it is important to provide lifestyle related interventions, such as resistance training, to improve the quality of life of older people living with diabetes.[37] At present there is limited information on the physical function of older people with diabetes compared with those without diabetes, as well as how acceptable or feasible a resistance training intervention would be in this group.

Resistance exercise training appears to be a promising intervention to improve the health of those living with insulin treated diabetes, particularly those who are older. However, we have a limited understanding of what form such a resistance training programme might take, and how and to what extent it will improve health.

### Aims and objectives

The purpose of this baseline case–control descriptive observational study and subsequent feasibility trial is to characterise the physical function, cardiovascular health, and the health and well-being of older people with mild frailty and with/without insulin-treated diabetes, and to test the feasibility of conducting a trial of resistance training in improving these parameters, and the acceptability of regular resistance exercise as a modality to improve health outcomes in older people with insulin treated diabetes.

## METHODS AND ANALYSIS
### Study design

This is a single-centre interventional feasibility randomised controlled trial with an associated baseline case–control descriptive observational component, and a qualitative and process evaluation component, conducted in Newcastle upon Tyne, England. Thirty participants with insulin treated diabetes and mild frailty, and thirty without diabetes will be recruited. All participants will be aged ≥60. All participants will undergo blood and physical testing, for the baseline case–control component. The diabetic participants will then go forward into the trial. They will be randomised 1:1 to the intervention group, which is a 4-week programme of supervised resistance exercise training, or to the control group; to carry on with any usual activity as normal. The current version of the protocol is V.3.

## Exclusion and inclusion criteria

Inclusion criteria for diabetes group (n=30):
- ► Adults ≥60 years.
- ► Type 1 diabetes OR type 2 diabetes treated with exogenous insulin.
- ► BMI <30 in participants with type 2 diabetes.
- ► Rockwood Clinical Frailty Score of 3 or 4.

Inclusion criteria for non-diabetes group (n=30):
- ► Adults ≥60 years.
- ► Rockwood Clinical Frailty Score of 3 or 4.

Exclusion criteria for all groups:
- ► History of myocardial infarction, stroke, renal failure, severe hypertension or liver disease in the last 12 months.
- ► Unsuitable for the intervention due to limiting musculoskeletal problems.
- ► Inability to give written informed consent.

## Identification, recruitment and consent procedures

All potential participants will be identified through the following methods: by their treating clinician who is a clinical member of the research team, in clinic at the Newcastle Diabetes Centre (applicable to those with diabetes only); via poster adverts in general practitioner (GP) practices and other secondary care clinics; GP practice database searches facilitated by the North East and North Cumbria Clinical Research Network; via social media; and via the Newcastle United Foundation charity. All methods have been reviewed and approved by the Health Research Authority and the study sponsor, through their ethical and governance review processes.

A participant information sheet will be sent to potential participants. Informed written consent will be given and eligibility confirmed by a member of the research team. Potential participants will then be screened with the Rockwood Clinical Frailty Score, by either their treating diabetes clinician (where applicable) or by a member of the research team. The study will take place from December 2020 to September 2022.

## Study procedures

### Initial procedures

All 60 participants will undergo the following blood/cardiovascular, physical tests, and patient reported outcome measures at the Newcastle Clinical Research Facility, at the start of the trial and after the 4-week intervention/control period:

### Blood and cardiovascular:

1. Resting blood pressure.
2. A 15 mL blood sample will be taken via venepuncture for the quantification of: glycated haemoglobin (HbA1c), blood lipid profile, inflammatory cytokines by routine hospital clinical chemistry or Newcastle Laboratories. 5 mL will be used to assess counts of endothelial progenitor cells, by flow cytometry as previously described, for a deeper investigation of vascular health in this patient group.[38]

### Physical function

1. Body composition: height, weight, waist circumference, % body fat and % fat free mass using bioelectrical impedance analysis (SECA 515 Body Composition Analyser).
2. Isometric strength: a torque and strain gauge will be used to assess the force capability of the participants' lower limbs. This test involves maximally extending the leg against an immovable strain gauge, this allows for the calculation of peak force and time-course changes in force.
3. Isometric strength: a torque and strain gauge will be used to assess the force capability of the participants' lower limbs. This test involves maximally extending the leg against an immovable strain gauge, this allows for the calculation of peak force and time-course changes in force.
4. Gait speed: using digital timing gates, the participants will be required to complete three 4 m walking tests, to assess the normal walking speed of the participants.
5. Timed Sit to stand: participants will sit on a chair and complete five stand and sit movements without use of the arms.

### Patient-reported outcome measures

For all participants:
1. Health-related quality of life: the Short Form-36.

For participants with diabetes only:
2. Problem Areas in Diabetes scale.
3. Hypo Fear Scale.

### Clinical history

For all participants, information on:
1. Comorbid disease.
2. Current medications, including changes in medications during the trial period.
3. Weight loss.
4. Exhaustion.
5. Physical activity levels (using the International Physical Activity Questionnaire, short form).

For participants with diabetes only, information on:
1. Insulin regimen.
2. Glucose monitoring (self-report).
3. Serious hypoglycaemic episodes over the past 12 months.

## Randomisation

After completing the initial testing procedures, the diabetic group (n=30) will be randomised in a 1:1 ratio to either the intervention group (4-week supervised resistance training programme), or the control group (see figure 1).

Randomisation of the diabetic group will take place at the end of the baseline visit. Randomisation will be done in a Good Clinical Practice (GCP)-compliant manner using a web-based randomisation system (http://www.randomization.com/). The allocation sequence will be prepared by individuals who will remain independent of

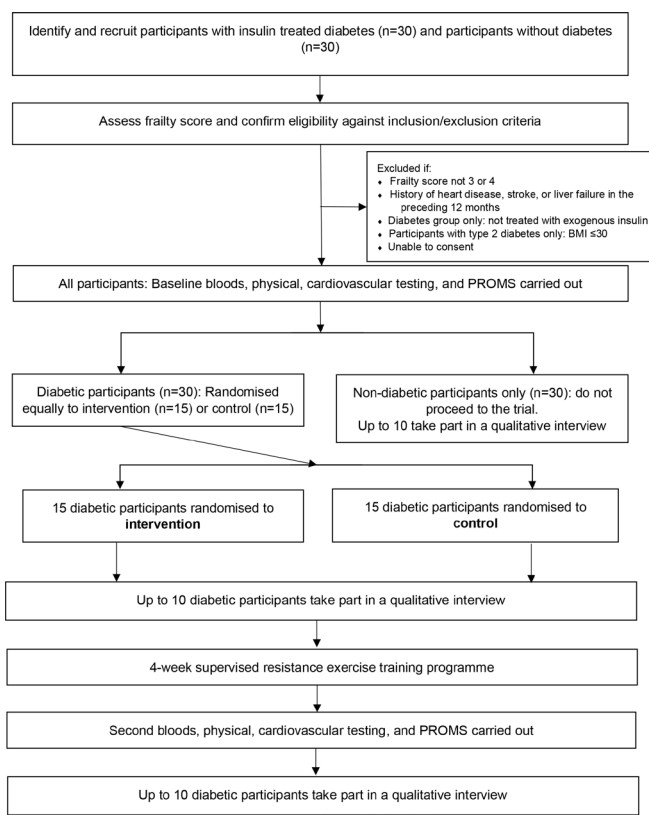

Identify and recruit participants with insulin treated diabetes (n=30) and participants without diabetes (n=30)

↓

Assess frailty score and confirm eligibility against inclusion/exclusion criteria

→

Excluded if:
• Frailty score not 3 or 4
• History of heart disease, stroke, or liver failure in the preceding 12 months
• Diabetes group only: not treated with exogenous insulin
• Participants with type 2 diabetes only: BMI ≤30
• Unable to consent

↓

All participants: Baseline bloods, physical, cardiovascular testing, and PROMS carried out

↓

Diabetic participants (n=30): Randomised equally to intervention (n=15) or control (n=15)

Non-diabetic participants only (n=30): do not proceed to the trial.
Up to 10 take part in a qualitative interview

↓

15 diabetic participants randomised to **intervention**

15 diabetic participants randomised to **control**

↓

Up to 10 diabetic participants take part in a qualitative interview

↓

4-week supervised resistance exercise training programme

↓

Second bloods, physical, cardiovascular testing, and PROMS carried out

↓

Up to 10 diabetic participants take part in a qualitative interview

**Figure 1** Consolidated Standards of Reporting Trials diagram. BMI, body mass index; PROMS, patient reported outcome measures.

the study team to preserve allocation concealment. The randomisation code sequence will not be accessible by the study team until after the trial analysis is complete.

## Intervention

The intervention is a 4-week, semistructured resistance exercise training programme, designed to increase muscle mass and strength. Training will be carried out at participants' preferred public gym, and facilitated by a trained member of the research team. The programme involves 2–3 sessions lasting less than 1 hour each, per week, for each of the 4 weeks. Weeks 1 and 2 will be fully supervised by a member of the research team. In week 3, participants will be asked to train alone in one of the sessions, and in week 4, they will train fully independently. A 4-week programme has been selected to assess feasibility and acceptability to participants, similar to previous feasibility work carried out in our team with older people.[36] The programme is not designed to induce changes in any physical or clinical outcomes.

The trial will be carried out once all relevant COVID-19 restrictions have been lifted in England. The research team will also adhere to COVID-19 standard operating protocols specified by the sponsor.

## Resistance exercise training programme design

Following extensive explanation and demonstration of proper exercise technique. For each exercise, resistance is increased until momentary failure occurs within 10

repetitions. One repetition maximum (1RM) is estimated using a prediction equation based on using the variables of 'load lifted' and 'number of repetitions completed'.[39] This method has been previously demonstrated as a valid approach for estimating 1RM in older people.[40]

The following exercise sessions will be completed weekly for 4 weeks:

### Session 1

Leg press, leg extension, leg curl, leg adduction, calf raises, chest press, shoulder press, lateral pull down, lateral raises.

Repetitions: 8–12 at 70% 1RM, Sets per exercise: 3, recovery between sets: 2 min.

### Session 2

Leg press, single-leg half leg press, chest press, shoulder press, seated row.

Repetitions: 5–8 at 85% 1RM, Sets per exercise: 3, recovery between sets: 4 min or feeling recovered.

### Session 3

Leg press, leg extension, leg curl, leg adduction, chest press, shoulder press, lateral pull down, lateral raises.

Repetitions: 12–15 at 60% 1RM, Sets per exercise: 3, recovery between sets: 2 min.

## Control arm

Participants randomised to the control arm will be asked to carry on with normal daily activities, without any changes to any exercise they might do.

## Blinding

Blinding will not be possible for participants as the intervention involves undertaking a supervised exercise resistance training programme, with the control arm undertaking no additional exercise other than any usual level of activity. The clinical team, and the research staff responsible for analysing quantitative outcomes, will be blinded to treatment allocation. Research staff responsible for supervising the resistance training, and the qualitative aspects of the study, will not be blinded.

## Qualitative process evaluation

We will use qualitative methods to develop an in-depth understanding of participant perceptions and experiences on the following topics: age and frailty, physical activity, living with diabetes (where relevant), barriers and facilitators to participating in the study and resistance training more generally, and views on the resistance training programme (where relevant). Prior to the intervention commencing, one semistructured interview will be conducted with up to 20 participants. These will be split equally between the intervention group and control group, and between those with and without diabetes. We will then interview up to 10 intervention group participants after the 4-week training programme, to understand their views and experiences of the programme and perceived impact on their health and well-being. We will

also offer participants the option of taking part in this interview as a one-off procedure, without taking part in the trial, should the trial be severely impacted by the COVID-19 global pandemic. We expect each interview to last 45–60 min. Interviews will take place at a time and location most suitable for the participants, either face to face or over the telephone. All participants will be approached for interview in order of recruitment. We anticipate that the sample sizes described will allow data saturation, and interviews will cease once no new semantic codes are identified from our concurrent thematic analysis of these data (code saturation).[41]

### Patient and public involvement

Involvement of the public and stakeholders in the early stages of this study confirmed that the potential impact of exercise on blood glucose control, especially hypoglycaemia, is a common concern for people living with diabetes. Study design was enhanced by capturing the views of several patients with insulin treated diabetes. They reflected on the relevance and importance of the study, study documentation and approach, and potential dissemination strategies for the public. Two PPI members are actively involved in this trial, influencing design and conduct. Their input will be supported according to National Institute for Health Research (NIHR) Centre for Engagement and Dissemination guidance.

### Study outcomes
#### Feasibility outcomes

The primary aim of this study is acceptability and feasibility of procedures for recruitment and retention, randomisation and adherence and fidelity to the resistance training programme intervention. We will use a traffic light approach, where green=proceed without modification; amber=proceed but with modification, red=unrealistic to proceed without major modification. Recruitment rates will be calculated as the rate of invited participants who are eligible, who subsequently provide informed consent. Green ≥50% recruitment, amber=25%–50% recruitment, red ≤25% recruitment. Attrition rates will be measured, defined by discontinuation of the resistance training intervention and/or lost to follow-up measurement for both conditions. Green ≤10% attrition, amber=10%–20% attrition, red ≥20% attrition. Reasons for attrition will be explored qualitatively. Acceptance, adherence and fidelity to the resistance training intervention will be monitored by the research team, who will measure session attendance: green ≥90% attendance, amber=75%–90% attendance, red ≤75% attendance, plus participant following intervention instructions >75% of the time, and participant self-report. Preintervention and postintervention qualitative interviews will be used to assess the acceptability of the resistance training intervention, influences on diabetes self-management where applicable, and well-being more generally, at baseline and at 5 weeks. Information about adverse events will be collected for the intervention group.

### Secondary outcomes
#### Clinical and physical outcomes

1. Body composition measured using height, weight, waist circumference, percentage body fat and percentage fat-free mass using bioelectrical impedance analysis at baseline and 5 weeks.
2. Isometric strength measured using a torque and strain gauge at baseline and 5 weeks.
3. Handgrip strength measured using a digital handgrip dynamometer at baseline and 5 weeks.
4. Gait speed measured using three 4 m walking tests on digital timing gates at baseline and 5 weeks.
5. Timed sit to stand, measured using five sit-to-stand movements at baseline and 5 weeks.
6. Cardiovascular health measured using resting blood pressure, HbA1c, blood lipid profile, and inflammatory cytokines at baseline and 5 weeks.
7. Instances of hypoglycaemia, measured weekly from baseline, obtained using patient self-report.

### Sample size calculation

This is a feasibility study with no existing data to draw on to inform a meaningful sample size calculation. We have selected a sample size in line with previous guidance on feasibility studies,[42] and data on key outcomes collected during this trial will inform the sample size calculation for a larger efficacy trial in future.

### Data collection and management

Data will be collected by DJW, RS and GST, members of the research team. Quantitative data on blood, cardiovascular, and physical function tests will be gathered using a tailored case report form. Qualitative data, including non-participant observations of the training programme, will be recorded with a voice recording device alongside written field notes.

### Data analysis plan
#### Quantitative analysis

This is a feasibility study to inform a larger trial and no hypothesis testing will be conducted. Consequently, quantitative data analysis will be descriptive. Mean, SD, range and 95% CI will be assessed on all quantitative data to assess response rates, numbers of individuals consented and randomised, retention rate, fidelity to the intervention, and participation in the training programme and qualitative interviews. The same descriptive methods will be used to report questionnaire and assessment data at baseline and 5 weeks. Statistical analyses will be conducted using IBM SPSS Statistics V.22 software.

#### Qualitative analysis

Qualitative data (generated by interviews) will be analysed for thematic content. This approach is both inductive (data interrogated to answer research questions but themes allowed to 'emerge' from the data) and iterative (data collection and analysis occurring simultaneously). All interviews will be audio recorded and transcribed verbatim. Data analysis will involve a process of organising

the data, descriptive coding, interpretive coding, writing and theorising. Data will be managed using a qualitative computer software package (NVivo V.11).

Initially we will seek to understand each participant group (intervention and control), then we will explore similarities and differences across each group. Throughout this process, the constant comparative method of analysis will be used, with an iterative process of data collection and analysis. This will allow identification of initial themes and ideas from the data to be explored in more depth in subsequent interviews, and allows data from different participants to be compared and contrasted, such as intervention vs control participants, participants with type 1 or 2 diabetes, with different levels of frailty and so on. Deviant cases will be actively sought throughout the analysis, and emerging ideas and themes modified in response.

## Monitoring and trial management

A data monitoring committee has not been convened given the small size and feasibility focus of this trial. The Trial Management Group will provide trial oversight and monitor any safety issues that arise.

## Ethics and dissemination

Ethical approval was obtained from the UK Health Research Authority (ref: 20/NE/0178, North East - Newcastle and North Tyneside 2 Research Ethics Committee). This trial is sponsored by the Newcastle-upon-Tyne Hospitals NHS Foundation Trust (ref: 9144).

The report from the clinical trial will be used for publication and oral presentation at scientific meetings. The trial investigators aim to publish the results in writing in clinically relevant open access journals. A summary of findings will also be distributed to our patient and public involvement group.

## DISCUSSION

The health benefits of regular physical exercise for those with, and without, insulin treated diabetes are numerous. Many with diabetes choose not to participate in physical activity, often due to fear of hypoglycaemia, or limited knowledge of exercise types and regimens.[43] There is emerging evidence that resistance exercise can be doubly beneficial to older people with diabetes: it is an exercise modality which appears to carry less risk of hypoglycaemia than other forms of exercise, and has the potential to limit age-related physical deterioration exacerbated by diabetes, such as sarcopenia. However, we have a limited understanding of what form such a resistance training programme might take, how and to what extent it will improve health, how it might impact hypoglycaemia and how outcomes might differ between those with and without diabetes. In the EXPLODE study, we will test the acceptability of one resistance exercise training programme, and gather data on how the programme influences the health of older people with insulin treated diabetes. Should this feasibility study generate positive data and demonstrate participant acceptability, we intend to carry out a pilot randomised controlled trial of the same resistance training intervention over a longer duration.

In summary, EXPLODE is a single-centre feasibility randomised parallel group trial investigating whether resistance exercise training has the potential to improve the health of older people living with insulin treated diabetes. It will also provide us with information on the acceptability of the resistance training programme and any required design amendments to a future larger pilot.

**Contributors** RS contributed to research design and drafted the manuscript. DW, JS, GT and MW contributed to research design and revision of the manuscript. All authors approved the final version of the manuscript to be published. DW is responsible for the integrity of the work as a whole.

**Funding** This study is being supported by the Wellcome Trust, by a Wellcome Trust Small Grant (grant number: N/A) to DW.

**Disclaimer** The funder (Wellcome Trust) and sponsor (Newcastle upon Tyne Hospitals NHS Foundation Trust) will have no role in the study design, conduct, data analysis, results interpretation, or writing.

**Competing interests** None declared.

**Patient and public involvement** Patients and/or the public were involved in the design, or conduct, or reporting, or dissemination plans of this research. Refer to the Methods section for further details.

**Patient consent for publication** Not applicable.

**Provenance and peer review** Not commissioned; externally peer reviewed.

**ORCID iDs**
Rachel Stocker http://orcid.org/0000-0002-8189-2746
Miles D Witham http://orcid.org/0000-0002-1967-0990

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
