## [Reviewer comments · BMJ Open]

ARTICLE DETAILS

TITLE (PROVISIONAL)	EXercise to Prevent frailty and Loss Of independence in insulin treated older people with DiabetEs (EXPLODE): protocol for a feasibility randomised controlled trial (RCT)
AUTHORS	Stocker, Rachel; Shaw, James; Taylor, Guy S; Witham, Miles; West, Daniel J

VERSION 1 – REVIEW

REVIEWER	Ahmed Abdelhafiz Rotherham General Hospital, Geriatric Medicine
REVIEW RETURNED	22-Mar-2021

GENERAL COMMENTS	Important clinical topic that hardly been addressed in literature.
--

REVIEWER	Lindsay Nagamatsu Western University, Kinesiology
REVIEW RETURNED	28-Apr-2021

GENERAL COMMENTS	This protocol paper provides details on a feasibility randomized controlled trial aimed at examining the feasibility of recruiting and engaging older adults with diabetes in a resistance training program aimed at improving physical functioning. The authors plan to recruit 30 older adults with diabetes and 30 matched controls (non-diabetics). Participants will be randomized to either a 4 week supervised resistance training program or a no-contact control group. This protocol is well written and clear. I have a few clarification questions that would strengthen the protocol. Methods:  -For the inclusion criteria, why are those with a BMI > 30 excluded in those with T2D? -There is no mention of physical activity level in the inclusion or exclusion criteria. Might current or prior experience with regular exercise (and resistance training) impact the results of the study? -How will participants for the qualitative interviews be selected? -Given that this is a feasibility study, how will feasibility be determined? The primary outcomes include recruitment and retention rates, adherence, etc. But is there a certain cutoff that would render the trial "feasible"? I.e., based on the data collected, how will the authors determine whether a full-scale trial is feasible or not? -I question whether the non-diabetic group is necessary given the aims of this feasibility study. What is the purpose of the non-diabetic group, how will the 2 groups (diabetic vs. non-diabetic) be compared, and could the authors achieve their main objectives without this group?
---

	Minor: -Page 10, line 25 – “compared” instead of “compared”
REVIEWER	Nitha Joseph University of Texas Health Science Center at Houston
REVIEW RETURNED	26-Jul-2021
GENERAL COMMENTS	BMI: specify why that BMI cut off is selected? Social media advertisement and recruitment ethical and legal implications can be added Other comorbidities or confounding factors need to be addressed as previous like stroke can impact their strength training. Or those can be exclusion criteria or can be included as confounding factors in quantitative analysis.
REVIEWER	Natalia Ricci UNICID
REVIEW RETURNED	30-Jul-2021
GENERAL COMMENTS	Manuscript ID: bmjopen-2021-048932 EXercise to Prevent frailty and Loss Of independence in insulin treated older people with DiabetEs (EXPLODE): protocol for a feasibility randomised controlled trial (RCT) Although the topic is very interesting (diabetes, frailty and resistance exercises) and of high relevance, it is not clear the main study design of this project. It is ok to have a mixed methods, however here we have so many methodologies that it is confusing.  - case- control (comparison with non-diabetes) - RCT (resistance training) - qualitative (interviews) - process evaluation (steps to conduct the trial) Abstract Please avoid to use sentences that need citation, like “There are 3.9m people in the UK with diabetes.” Avoid the use of the word “elderly”. The objectives (in the abstract) did not match with the analysis and with the aims in the full text: 1) The comparison with non-diabetes 2) Only at the end of the abstract it is explained that qualitative data will be collected. 3) The efficacy will not be evaluated, this is stated in main text. Strengths and limitations of this study The second bullet point is a limitation, therefore the authors should first point out the strengths and then after the limitations. Introduction The introduction is well written. However it lacks an important feature for feasibility RCT studies proposed by the CONSORT “Scientific background and explanation of rationale for future definitive trial”. Aims Mainly describe a case-control (part 1) and a RCT (part 2). The

	authors will not evaluate the effectiveness, so why this is a aim? For part 1, you do not need a RCT design and not a feasibility study. Lacks the most important part of the feasibility study- the process evaluation and qualitative. Methods The authors should clarify each one of the methodologies that they will use. The eligibility criteria has many flaws. - What about cognitive impairment? - What about neuropathic problems that are common in diabetes patients? - The practice of other physical activity should be controlled. It was not clear how non-diabetes older adults will be recruited. It is not clear the process of randomization together with a age, gender and frailty matched control. How this process will be performed? There is no information about allocation concealment mechanism. How the pandemic will impact the trial is not clear. An important outcome measure is missing, a questionnaire or scale of independence of daily living. The title of the article highlighted the "Loss Of independence", but no measure is included. How physical activity level will be measured? Convenient public gym, how this will work? All public gyms have materials, and instructors trained for the trial? What you mean by short sessions? We know that a 4-week program is not enough for changes in older adults (specially mild frailty), and it is not clear how long the authors are planning to extend it for the real trial. How will be deal safety issues during the sessions, specially the unsupervised ones? The qualitative part is lacking rigours, the sample size cannot be inferred a priori. It will be interesting to interview those eligible but not willing to participate too. It is very different to have a face-to-face, or by phone interview. The use of on-line interviews seeing each other is better in the impossibility of a face-to-face. A time line with the study designs, measures, and others will help a lot to better understand all the features of this project.
--	---

REVIEWER	Javier Courel-Ibáñez University of Murcia, Faculty of Sport Sciences
REVIEW RETURNED	07-Aug-2022

GENERAL COMMENTS	This is a nice RCT which could be a critical contribution to the existing literature on exercise, ageing and diabetes. I read the paper with interest and I have just some minor suggestions that I hope you find of interest.
--

	Inclusion/Exclusion criteria: - After checking the published protocol (ISRCTN13193281) I find the authors adds an inclusion criterion “BMI <30 in participants with type 2 diabetes”. Please explain briefly the rationale of this threshold. Measurements: - Probably the trial will be benefit from more upper-limbs tests as only handgrip is present and might not be properly explaining the changes after the intervention in frail older adults (https://pubmed.ncbi.nlm.nih.gov/24903908/). I suggest including a more functional tests such as estimate 1RM test for bench press exercise. Intervention. - “One repetition maximum (1RM) is estimated using a prediction equation based on using the variables of ‘load lifted’ and ‘number of repetitions completed” While this is a traditional approach (1993, 1999 references), current updated resistance training methods are benefited from the use of technology to accurately estimate the load and intensity. An example is the Velocity-Based Resistance Training (plase check: https://journals.lww.com/nsca-scj/Fulltext/2021/04000/Velocity_Based_Training__From_Theory_to.4.aspx). Lately, this approach has been successfully implemented among older adults (https://peerj.com/articles/7533/, https://pubmed.ncbi.nlm.nih.gov/33080817/). If possible, I would suggest the authors to incorporate this approach to collect velocity data, not only for exercise prescription purposes but also to enlarged the list of dependent variables (i.e., compare whether the velocities attained against a given load increases after the intervention). - “For each exercise, resistance is increased until momentary failure occurs within 10 repetitions.” Again, despite this is an accepted, traditional approach, latest recommendations favours resistance training not to failure (https://pubmed.ncbi.nlm.nih.gov/33555822/), even in older adults (https://link.springer.com/article/10.1007%2Fs12603-021-1665-8). Besides, explosive muscle actions must be included and emphasized within the regime: “Optimal training regimens for maximising muscle power should be performed with the concentric (shortening) phase as fast as possible, followed by a controlled, slower eccentric (lengthening) phase, focused on the lower limbs (27, 87). Sets of explosive muscle actions can be performed alone (69, 88) or combined with traditional resistance training during the same session, but always avoiding concentric failure (87, 89, 90).” If possible, I would suggest authors to adapt the intervention according to the latest evidence. Finally, one typo: P13, L26: “....insulin”
--	---

VERSION 1 – AUTHOR RESPONSE

Reviewer 1.

Dr. Ahmed Abdelhafiz, Rotherham General Hospital

Comment raised	Response by author
Important clinical topic that hardly been addressed in literature.	Thank you for your supportive comment.

Reviewer 2.
 Dr. Lindsay Nagamatsu

This protocol paper provides details on a feasibility randomized controlled trial aimed at examining the feasibility of recruiting and engaging older adults with diabetes in a resistance training program aimed at improving physical functioning. The authors plan to recruit 30 older adults with diabetes and 30 matched controls (non-diabetics). Participants will be randomized to either a 4 week supervised resistance training program or a no-contact control group. This protocol is well written and clear. I have a few clarification questions that would strengthen the protocol.

Comment raised	Response by author
-For the inclusion criteria, why are those with a BMI > 30 excluded in those with T2D?	Established insulin-treated non-obese type 2 diabetes shares many characteristics with type 1 diabetes, due to relatively greater insulin deficiency and lower insulin resistance than in type 2 diabetes associated with obesity. This includes intrinsic glucose variability with higher risk of impaired awareness of hypoglycaemia^{1 2}, including severe events requiring assistance from others in treatment.³ We hypothesise that mild frailty may have a comparable impact in type 1 diabetes and insulin-treated type 2 diabetes where BMI is <30 kg/m²⁴, with potentially comparable impacts of resistance exercise training. These are important challenges for both the older type 1 and 2 diabetes individual. We have added this information in the Introduction (page 5 lines 14-20) to clarify and provide a scientific rationale for our choice.
-There is no mention of physical activity level in the inclusion or exclusion criteria. Might current or prior experience with regular exercise (and resistance training) impact the results of the study?	We will be collecting information on physical activity levels and will use the information to inform later study design (as this is feasibility work and not a pilot or RCT).
-How will participants for the qualitative interviews be selected?	We have added additional information about our selection approach and data saturation (p14 line 18-21). We will approach all participants in order of recruitment to ensure inclusivity. Interviews will be conducted until we achieve saturation of data, i.e. no more new semantic codes are being identified.
-Given that this is a feasibility study, how will feasibility be determined? The primary outcomes include recruitment and retention rates, adherence, etc. But is there a certain cutoff that would render the trial “feasible”? I.e., based on the data collected, how will the authors determine whether a full-scale trial is feasible or not?	Feasibility outcomes and their measurement (where appropriate) are described and have been further clarified in the ‘study outcomes’ section (p15 lines 11-23, p16 lines 1-21). We have also added our approach to assessing feasibility, using a traffic light system with associated cut-offs for feasibility aspects of the trial, on page 15 lines 14-23; page 16 lines 1-2.

-I question whether the non-diabetic group is necessary given the aims of this feasibility study. What is the purpose of the non-diabetic group, how will the 2 groups (diabetic vs. non-diabetic) be compared, and could the authors achieve their main objectives without this group?	Thank you for your helpful comment, which provoked constructive discussions within the study team. As you highlighted, we recognise that the non-diabetic group is not necessary for the trial itself. We have decided to remove the non-diabetic group from the trial. Our original reasoning for their inclusion was to allow us to identify non-diabetic related, and diabetic related, issues relating to exercise in this group. However this can be better achieved by including the non-diabetic group in a baseline case-control study, occurring immediately prior to the trial itself. Our design is now as follows:  1) a baseline case-control descriptive observational study, with 30 diabetics and 30 without (all aged 60 or over with mild frailty). This is to gather data on physical status, allowing for a comparison between diabetics and non-diabetics. 2) a feasibility RCT involving the 30 diabetic participants only. Once 1) is complete, they will be randomised 1:1 into the intervention group (n=15) and control group (n=15). We will not carry out any age/sex/frailty matching.
Minor: -Page 10, line 25 – “compared” instead of “compared”	Thank you – amended.

Reviewer: 3

Dr. Nitha Joseph, University of Texas Health Science Center at Houston

Comment raised	Response by author
BMI: specify why that BMI cut off is selected?	Established insulin-treated non-obese type 2 diabetes shares many characteristics with type 1 diabetes, due to relatively greater insulin deficiency and lower insulin resistance than in type 2 diabetes associated with obesity. This includes intrinsic glucose variability with higher risk of impaired awareness of hypoglycaemia^{1 2}, including severe events requiring assistance from others in treatment.³ We hypothesise that mild frailty may have a comparable impact in type 1 diabetes and insulin-treated type 2 diabetes where BMI is <30 kg/m²⁴. with potentially comparable impacts of resistance exercise training.

	These are important challenges for both the older type 1 and 2 diabetes individual. We have added this information in the Introduction (page 5 lines 14-20) to clarify and provide a scientific rationale for our choice.
Social media advertisement and recruitment ethical and legal implications can be added	We have added that all advertisement methods have been reviewed and approved by the sponsor and the Health Research Authority ethical and governance committees/processes (page 8 lines 18-19).
Other comorbidities or confounding factors need to be addressed as previous like stroke can impact their strength training. Or those can be exclusion criteria or can be included as confounding factors in quantitative analysis.	We have now made clear in the exclusion criteria that anyone with a history of stroke in the past 12 months will be excluded. (page 8 lines 5-9)

Reviewer: 4
Dr. Natalia Ricci, UNICID

Comment raised	Response by author
Although the topic is very interesting (diabetes, frailty and resistance exercises) and of high relevance, it is not clear the main study design of this project. It is ok to have a mixed methods, however here we have so many methodologies that it is confusing.  - case-control (comparison with non-diabetes) - RCT (resistance training) - qualitative (interviews) - process evaluation (steps to conduct the trial) 	Thank you for your helpful comment, which provoked constructive discussions within the study team. As you highlighted, we recognise that the non-diabetic group is not necessary for the trial itself. We have decided to remove the non-diabetic group from the trial. Our original reasoning for their inclusion was to allow us to identify non-diabetic related, and diabetic related, issues relating to exercise in this group. However this can be better achieved by including the non-diabetic group in a baseline case-control study, occurring immediately prior to the trial itself. Our design is now as follows:  1) a baseline case-control descriptive observational study, with 30 diabetics and 30 without (all aged 60 or over with mild frailty). This is to gather data on physical status, allowing for a comparison between diabetics and non-diabetics. 2) a feasibility RCT involving the 30 diabetic participants only. Once 1) is complete, they will be randomised 1:1 into the intervention group (n=15) and control group (n=15). We will not carry out any age/sex/frailty matching.
Abstract Please avoid to use sentences that need citation,	We believe that this statistic is relevant for inclusion in the paper, as it demonstrates the

like “There are 3.9m people in the UK with diabetes.”	scale of the population and resulting clinical concern. We also reiterate this in the opening sentence with appropriate citation.
Avoid the use o the word “elderly”.	Changed to ‘older people’ throughout the manuscript.
The objectives (in the abstract) did not match with the analysis and with the aims in the full text: 1) The comparision with non-diabetes 2) Only at the end of the abstract it is explained that qualitative data will be collected. 3) The efficacy will not be evaluated, this is stated in main text.	We have added ‘with/without insulin treated diabetes’ to aim (1) (page 1 line 20), to clarify that we are recruiting and comparing those with diabetes and without diabetes in the baseline case-control study. In aim (2) we have amended ‘test’ to ‘understand’... the feasibility and acceptability of a four-week resistance exercise training programme. (page 1 line 20-21).This is to better capture the fact that we will be carrying out qualitative data collection. Efficacy has been reworded to acceptability – this better describes the aim of this feasibility trial.
Strengths and limitations of this study The second bullet point is a limitation, therefore the authors should first point out the strengths and them after the limitations.	We have revised the strengths and limitations section in line with your comments. (page 3 lines 2-14).
Introduction The introduction is well written. However it lacks an important feature for feasibility RCT studies proposed by the CONSORT “Scientific background and explanation of rationale for future definitive trial”.	Thank you for your comment. We have now made clear in our aims and objectives section the particular aspects we are investigating to inform future definitive trial design.
Aims Mainly describe a case-control (part 1) and a RCT (part 2). The authors will not evaluate the effectiveness, so why this is a aim? For part 1, you do not need a RCT design and not a fesibility study. Lacks the most important part of the feasiability study- the process evaluation and qualitative.	We have amended ‘efficacy’ to ‘acceptability’ to better describe our aims, and to illustrate that this encompasses qualitative data collection in addition to quantitative. We have also amended our design, please see page 7 lines 8-16 (and our response to your first comment).
Methods The authors should clarify each one of the methodologies that they will use.	We have amended our design, please see page 7 lines 8-16 (and our response to your first comment). Also, we have clarified that our RCT will include qualitative and process evaluation components.
The elegibility criteria has many flaws. - What about cognitive impairment? - What about neuropatic problems that are common in diabets patients? - The practice of other physical activity should be controled.	Those with cognitive impairment which will impact informed consent processes will be excluded as per the final exclusion criterion. However, mild degrees of cognitive impairment do not necessarily preclude giving informed consent, and enabling inclusion of those with

	mild cognitive impairment increases the generalisability of the findings. We do not intend to exclude those with neuropathy as this in itself does not always limit engagement in physical activity. And omitting these people would limit the generalisability of the findings, which we are keen to avoid. As this is a feasibility study we do not intend to control existing physical activity. We will take information on existing physical activity levels (as per 'clinical history, e') . We will consider controlling physical activity levels in a future definitive trial, if necessary. We are using the Rockwood Clinical Frailty Scale to further ensure that only those with a very modest level of frailty will be identified.
It was not clear how non-diabetes older adults will be recruited.	We have added 'all' to 'potential participants' at the start of the 'identification, recruitment, and consent procedures' section (p8 line 13). This clarifies that all recruitment methods, except the diabetes clinic, apply to both the diabetic and non-diabetic group.
It is not clear the process of randomization together with a age, gender and frailty matched control. How this process will be performed?	We have updated our description of the randomisation process on page 11 lines 10-21. In line with the amendment to the study design, we are now not matching age/gender/frailty.
There is no information about allocation concealment mechanism.	We have updated our description of the allocation concealment mechanism on page 11 lines 18-21.
How the pandemic will impact the trial is not clear.	We have added Covid-19 related information in the 'intervention' section, page 12 lines 11-13.
An important outcome measure is missing, a questionnaire or scale of independence of daily living. The title of the article highlighted the "Loss Of independence", but no measure is included.	The maintenance of independence is the aim of a future definitive RCT – this feasibility study is the foundations of this future work. The outcomes chosen for this trial are to explore signals of activity of the intervention (on physical performance and cardiometabolic parameters), and to describe the baseline characteristics of the trial cohort in some detail.
How physical activity level will be measured?	By participant self-report, in minutes, using the International Physical Activity Questionnaire (short form). We have added this detail on page 10 line 23.
Convenient public gym, how this will work? All public gyms have materials, and instructors trained for the trial?	Changed wording to 'preferred' (page 12 line 3) to clarify that participants have their own choice of gym.

	Added 'facilitated by a trained member of the research team' (page 12 line 3) to clarify that the research team are facilitating sessions, i.e. acting as an instructor.
What you mean by short sessions?	Amended to 'sessions lasting less than one hour each'. (page 12 line 4)
We know that a 4-week program is not enough for changes in older adults (specially mild frailty), and it is not clear how long the authors are planning to extend it for the real trial.	The programme is not designed to induce changes in any physical or clinical outcomes – only to assess feasibility and acceptability. The acceptability data we gather will inform programme length for the definitive trial.
How will be deal safety issues during the sessions, specially the unsupervised ones?	All sessions will be monitored. The supervised sessions will be monitored for safety by the member of the research team acting as instructor. Gym staff will monitor participants during unsupervised sessions, as part of their normal working role at the gym.
The qualitative part is lacking rigours, the sample size cannot be infered a prior. It will be interesting to interview those elegible but not willing to participate too.	We have added further information about our sampling strategy and data saturation to the relevant sections of the manuscript. (p14 line 18-21).
It is very different to have a face-to-face, or by phone interview. The use of on-line interviews seeing each other is better in the impossibility of a face-to-face.	We agree. If face-to-face interviews are impossible, our preferred method is video calling rather than an audio only call.
A time line with the study designs, measures, and others will help a lot to better understand all the features of this project.	We have further clarified the flow of the project in the manuscript. For full flow of procedures please see Figure 1.

Reviewer: 5

Prof. Javier Courel-Ibáñez, University of Murcia

This is a nice RCT which could be a critical contribution to the existing literature on exercise, ageing and diabetes. I read the paper with interest and I have just some minor suggestions that I hope you find of interest.

Comment raised	Response by author
Inclusion/Exclusion criteria: - After checking the published protocol (ISRCTN13193281) I find the authors adds an inclusion criterion "BMI <30 in participants with type 2 diabetes". Please explain briefly the rationale of this threshold.	Established insulin-treated non-obese type 2 diabetes shares many characteristics with type 1 diabetes, due to relatively greater insulin deficiency and lower insulin resistance than in type 2 diabetes associated with obesity. This includes intrinsic glucose variability with higher risk of impaired awareness of hypoglycaemia ^{1 2} , including severe events requiring assistance from others in treatment. ³ We hypothesise that mild frailty may have a comparable impact in

	type 1 diabetes and insulin-treated type 2 diabetes where BMI is $<30 \text{ kg/m}^2$ with potentially comparable impacts of resistance exercise training. These are important challenges for both the older type 1 and 2 diabetes individual. We have added this information in the Introduction (page 5 lines 14-20) to clarify and provide a scientific rationale for our choice.
Measurements: - Probably the trial will be benefit from more upper-limbs tests as only handgrip is present and might not be properly explaining the changes after the intervention in frail older adults (https://pubmed.ncbi.nlm.nih.gov/24903908/) I suggest including a more functional tests such as estimate 1RM test for bench press exercise.	Thank you for your comments. This is a good suggestion, while we know handgrip strength is a predictor of various frailty outcomes, changes in other strength outcomes may be useful to collect. We will look to include this in the next phase of our project. However, it is important to note that we will monitor the training loads people are using during the training which should also indirectly track changes in functional strength in various upper body strength.
Intervention. - “One repetition maximum (1RM) is estimated using a prediction equation based on using the variables of ‘load lifted’ and ‘number of repetitions completed” While this is a traditional approach (1993, 1999 references), current updated resistance training methods are benefited from the use of technology to accurately estimate the load and intensity. An example is the Velocity-Based Resistance Training (please check: https://journals.lww.com/nsca-sci/Fulltext/2021/04000/Velocity_Based_Training_From_Theory_to.4.aspx) Lately, this approach has been successfully implemented among older adults https://peerj.com/articles/7533/ ; https://pubmed.ncbi.nlm.nih.gov/33080817/ If possible, I would suggest the authors to incorporate this approach to collect velocity data, not only for exercise prescription purposes but also to enlarged the list of dependent variables (i.e., compare whether the velocities attained against a given load increases after the intervention).	Thank you for these suggestions, our approach has been largely driven by experience of conducting exercise research in aging populations, led by Prof Witham at the Institute for Ageing. Moreover, we have found the implementation of our measures of frailty to be easily conducted in clinical settings without the need for specialist equipment (e.g. a non-exercise specialist can conduct most of our measures in a clinic waiting room). We have, however added isometric strength of the lower limb and will access measurements such as peak force, time to peak force, force at 100 ms, 200 ms, rate of force development. Also, 5x sit to stand is a measure of velocity (aka power) in lower limb function. Lastly, with regards to upper body strength measurement – this is something we will potentially add in the future – our experience is that changes in lower body strength are most important to capture as this tends to transfer to functional tasks such as stair climbing and sit to stand.
- “For each exercise, resistance is increased until momentary failure occurs within 10 repetitions.” Again, despite this is an accepted, traditional approach, latest recommendations favours resistance training not to failure	Thank you for your comments. As described above, we have based our approach on prior work by our team, and we are mindful that we do not have data from an older diabetic population in order to include your suggestions at this

https://pubmed.ncbi.nlm.nih.gov/33555822/ even in older adults https://link.springer.com/article/10.1007%2Fs12603-021-1665-8 Besides, explosive muscle actions must be included and emphasized within the regime: “Optimal training regimens for maximising muscle power should be performed with the concentric (shortening) phase as fast as possible, followed by a controlled, slower eccentric (lengthening) phase, focused on the lower limbs (27, 87). Sets of explosive muscle actions can be performed alone (69, 88) or combined with traditional resistance training during the same session, but always avoiding concentric failure (87, 89, 90).” If possible, I would suggest authors to adapt the intervention according to the latest evidence.	feasibility stage of the work. During our qualitative capture we will include details on this part of the study. As you suggest, this may be something that requires changing in the next phase of our work.
Finally, one typo: P13, L26: “...insulin”	Thank you, amended.

References

1. Yun JS, Park YM, Han K, et al. Association between BMI and risk of severe hypoglycaemia in type 2 diabetes. *Diabetes Metab* 2019;45(1):19-25. doi: 10.1016/j.diabet.2018.03.006 [published Online First: 2018/04/06]
2. van Meijel LA, de Vegt F, Abbink EJ, et al. High prevalence of impaired awareness of hypoglycemia and severe hypoglycemia among people with insulin-treated type 2 diabetes: The Dutch Diabetes Pearl Cohort. *BMJ Open Diabetes Research & Care* 2020;8(1):e000935. doi: 10.1136/bmjdr-2019-000935
3. Horii T, Oikawa Y, Kunisada N, et al. Real-world risk of hypoglycemia-related hospitalization in Japanese patients with type 2 diabetes using SGLT2 inhibitors: a nationwide cohort study. *BMJ Open Diabetes Research and Care* 2020;8(2):e001856.
4. Izzo A, Massimino E, Riccardi G, et al. A Narrative Review on Sarcopenia in Type 2 Diabetes Mellitus: Prevalence and Associated Factors. *Nutrients* 2021;13(1) doi: 10.3390/nu13010183 [published Online First: 2021/01/14]

VERSION 2 – REVIEW

REVIEWER	Javier Courel-Ibáñez University of Murcia, Faculty of Sport Sciences
REVIEW RETURNED	17-Sep-2021
GENERAL COMMENTS	The authors have successfully addressed the main concerns. I wish them all the best in the ongoing of this interesting project.